# Multi-Group Tri-plane Based Local Occupancy Estimation for Object Grasping

## Abstract

This paper addresses the challenge of robotic grasping of general objects. Similar to prior research, the task reads a single-view 3D observation (i.e., point clouds) captured by a depth camera as input. Crucially, the success of object grasping highly demands a comprehensive understanding of the shape of objects within the scene. However, single-view observations often suffer from occlusions (including both self and inter-object occlusions), which lead to gaps in the point clouds, especially in complex cluttered scenes. This renders incomplete perception of the object shape and frequently causes failures or inaccurate pose estimation during object grasping. In this paper, we tackle this issue with an effective albeit simple solution, namely completing grasping-related scene regions through local occupancy prediction. Following prior practice, the proposed model first runs by proposing a number of most likely grasp points in the scene. Around each grasp point, a module is designed to infer any voxel in its neighborhood to be either void or occupied by some object. Importantly, the occupancy map is inferred by fusing both local and global cues. We implement a multi-group tri-plane scheme for efficiently aggregating long-distance contextual information. The model further estimates 6-D grasp poses utilizing the local occupancy-enhanced object shape information and returns the top-ranked grasp proposal. Comprehensive experiments on both the large-scale GraspNet-1Billion benchmark and real robotic arm demonstrate that the proposed method can effectively complete the unobserved parts in cluttered and occluded scenes. Benefiting from the occupancy-enhanced feature, our model clearly outstrips other competing methods under various performance metrics such as grasping average precision.

## 1 Introduction

General object grasping (Fang et al., 2020; Wang et al., 2021; Liu et al., 2022; Xu et al., 2023) plays a critical role in a variety of robotic applications, such as manipulation, assembling and picking. Its success lies in the ability to generate accurate grasp poses from visual observations, without requiring prior knowledge of the scene's exact structure. In recent years, substantial advancements have occurred in this domain, leading to the widespread adoption of object grasping in both industrial and service sectors. Most of modern object grasping methods rely on either point clouds (Fang et al., 2020), RGB-D multi-modal images (Mahler et al., 2017) and voxels (Breyer et al., 2020)) as inputs for predicting the best grasp point / pose. To attain high success rate of grasping operations, it is crucial to have a full perception of the object shapes locally around each proposed grasp point. Nevertheless, since most methods only exploit a single snapshot of the target scene. The desired shape information is often incomplete owing to self-occlusion under specific camera viewpoint or mutual occlusion across adjacent objects. This causes lots of crucial volumtric clue unavailable when conducting object grasping, and thus leads to various failing cases. Figure 1 shows an illustrative case where in a failure the gripper collides with a target object (the red hair dryer) due to incompletely-estimated object shapes under the single-view setting.

There are multiple potential solutions to resolve the aforementioned issue, including the adoption of multi-view scene snapshots or multi-modal learning (*e.g.*, the joint optimization over color image and point clouds). This paper tackles this challenge by adhering to a depth-based scene representation in a single view, recognizing the practical limitations often associated with employing multi-view or multi-modal data. For example, in many cases the robotic arm is required to conduct

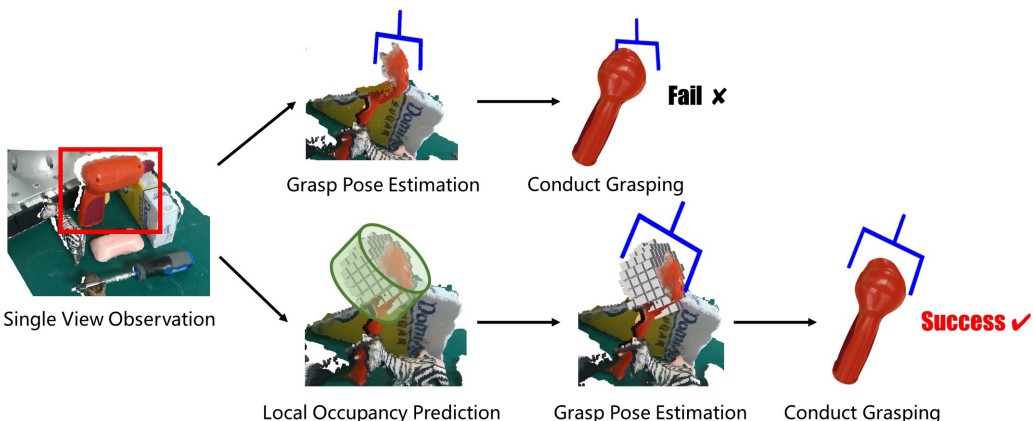

Figure 1: Illustration of local occupancy-enhanced object grasping. *Top*: The gaps in point clouds heavily affect the accuracy of estimated grasp poses, leading to a failure under self-occlusion in this case. *Bottom*: In our proposed method, grasp pose is estimated using local occupancy-enhanced features, which essentially infer the complete object shape locally around the grasp point by fusing global / local cues.

swift grasp estimation or physically constrained to observe the scene from a restricted set of viewing angles. To restore the occluded or unobserved portions of the objects, it is a natural idea to resort to some point cloud completion works (Yuan et al., 2018; Xia et al., 2021; Jiang et al., 2023; Gong et al., 2021). However, most of them focused on the fine-grained object-level completion and are compute-intensive. We instead adopt a voxel-based, relatively coarse-grained scene representation, and formulate the problem of object shape completion as inference over occupancy maps.

In this work we develop a local occupancy-enhanced object grasping method utilizing multi-group tri-planes. There are a vast literature on occupancy estimation neural networks (Song et al., 2017; Peng et al., 2020; Zhang et al., 2023) that recover the voxel-level occupancy of the scene from a single-view RGB / RGB-D image or point cloud. We argue that directly deploying these models are not computationally optimal for the task of object grasping, since most of them operate over the full scene and are not scalable to large scenes. There are two key considerations in expediting and improving the occupancy estimation. First, this work follows previous practice in object grasping that first generates a sparse set of most confident 3-D grasp points for a scene. For obtaining the optimal pose of the gripper, the geometry around current grasp point is centrally informative. The compute-demanding occupancy estimation is restricted to be within some local neighborhood of the grasp point, striking an accuracy / efficacy balance. Secondly, holistic scene context plays a pivotal role for precisely inferring the state of each voxel. However, learning over 3-D volumes is neither computationally feasible (the large number of voxels is not amenable to intensive convolutions or attention-based operations) nor necessary (most voxels are void and should not been involded in the computation). To effectively aggregate multi-scale information, we propose an idea of multi-group tri-plane projection to extract shape context from point clouds. Each tri-plane constructs three feature planes by projecting the scene along its three coordinates, providing a compact representation of the scene. Features on each plane are obtained by aggregating the information from partial observation along an axis. Since such projection is information lossy, we utilize multi-group tri-planes that are uniformly drawn from **SO(3)** and collectively preserve the major scene structures via diverse snapshots. When predicting the occupancy of an arbitrary voxel, a feature querying scheme fusing global and local context is proposed. Both the tri-plane based occupancy estimation and grasp pose estimation are jointly learned through an end-to-end differetiable optimization.

## 2 RELATED WORK

**Object grasping**. Most of the object grasping methods (Jiang et al., 2011; Lenz et al., 2015; Mahler et al., 2017; Morrison et al., 2018; Liang et al., 2019; Mousavian et al., 2019) leverage RGB-D image or point cloud to extract shape features for estimating grasp poses in cluttered scenes. Among them, Jiang et al. (2011); Lenz et al. (2015); Asif et al. (2018) take RGB-D images as input to estimate

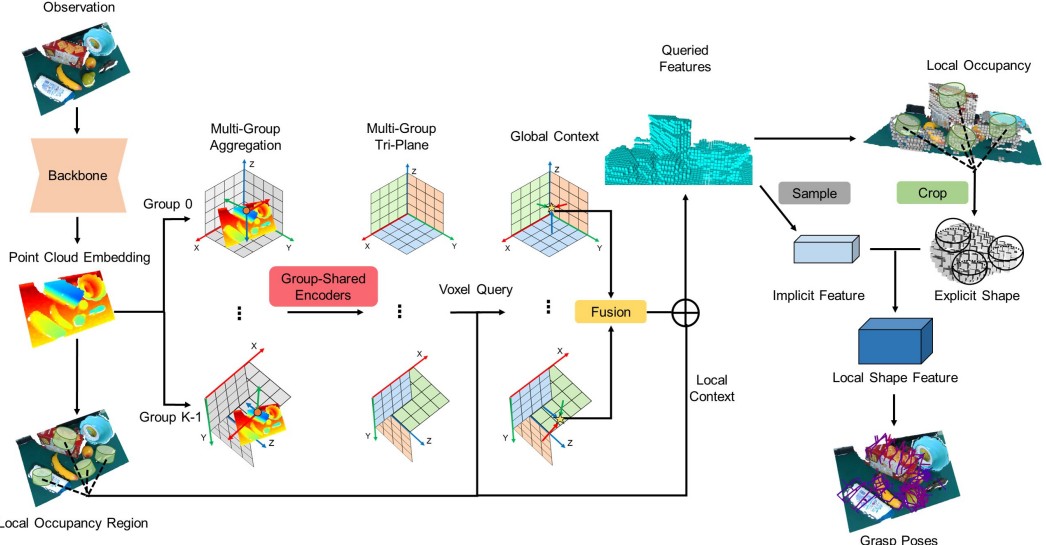

Figure 2: Model architecture of the proposed local occupancy-enhanced object grasping. It first identifies a number of interested local occupancy regions. Then multi-group tri-plane aggregates the scene context for local occupancy estimation. Finally the occupancy-enhanced local shape feature in each grasp region is extracted by fusing the information of both explicit voxels and implicit queried features, and is decoded to grasp poses.

grasp poses with rotated boxes. Liang et al. (2019); ten Pas et al. (2017); Ni et al. (2020); Qin et al. (2019) generate dense grasp poses and evaluate them to elect the best one. Fang et al. (2020) propose an end-to-end 6-D pose-estimating model from point clouds, and Wang et al. (2021) improve grasp point sampling via grasp-oriented affordance segmentation. Though several works (*e.g.*, Lundell et al. (2019; 2020); Varley et al. (2017)) have explored object completion to facilitate object grasping, these completions are essentially object-centric and not jointly optimized with grasping, preventing them from promoting grasp quality further and being applied to complex scenes.

**Occupancy network**. Several relevant networks (Mescheder et al., 2019; Cao & de Charette, 2022; Tian et al., 2023; Huang et al., 2023) have emerged recently to complete the scene from incomplete observations. Song et al. (2017) proposes a 3-D convolutional network in voxel form. Some works (Zhang et al., 2018; Cheng et al., 2020; Li et al., 2020) devise lighter and more expressive 3D convolutions for scene completion. Zhang et al. (2023); Li et al. (2023) propose to predict occupancy from RGB images through attentional modules. Huang et al. (2023) reduces the attention memory cost by tri-plane representation. Nevertheless, these methods need to predict occupancy densely for the whole scene, which is unnecessary and overburdened for grasping tasks.

**Tri-plane representation**. A few works (Chan et al., 2022; Gao et al., 2022; Noguchi et al., 2022; Skorokhodov et al., 2022; Shue et al., 2023; Anciukevicius et al., 2023) have utilized tri-plane as compact 3-D scene representation. Chan et al. (2022) and Peng et al. (2020) bring up the idea of representing a scene as three orthogonal feature planes for 3D rendering and occupancy prediction. Huang et al. (2023) uses the tri-plane representation for scene completion in outdoor autonomous driving tasks. Wang et al. (2023) improves the tri-plane performance by adding positional encoding to it. Hu et al. (2023) attempts to enrich the 3D information by lifting tri-plane to tri-volume. All of these methods only considered to aggregate context to a single group tri-plane. It will suffer from information loss in complex and occluded grasping scenes with limited views.

## 3 THE PROPOSED METHOD

This section elaborates on the proposed local occupancy-enhanced object grasping model in details. The model is fed with single-view point clouds and returns multiple optimal 6-D grasp poses. In specific, we use the grasp pose formulation in Fang et al. (2020), defining a 6-D grasp pose via grasp point, grasp direction, in-plane rotation, grasp depth and grasp width. Figure 2 illustrates the framework of our model.

### 3.1 LOCAL OCCUPANCY REGIONS

To begin with, we encode the input point cloud $\boldsymbol{P} \in \mathbb{R}^{3 \times N}$ with a 3-D UNet backbone network and obtain the point cloud embedding $\boldsymbol{F_P} \in \mathbb{R}^{C_P \times N}$, where $N$ and $C_P$ are the count of points and the number of feature channels respectively. Assume $G$ to be the collection of candidate grasp points. To construct $G$, we employ a grasp affordance segmentation procedure Wang et al. (2021) on observed point clouds, and sample a fixed number of possible grasp points (1,024 in our implementation) from the affordance area into $G$. For the grasp direction, we regress the view-wise affordance of each grasp point and choose the view with the maximum affordance as its grasp direction. Through these, areas with a large probability of having high quality grasp poses are selected to predict local occupancy and facility grasp pose estimation.

For a 2-fingered gripper, knowing its radius $r$ and grasp depth interval $[d_{min}, d_{max}]$, we define a local grasp region $S$ in the coordinate frame centered at the grasp point to be a cylinder reachable to the gripper $S = \{(x, y, z) \mid x^2 + y^2 \leq r^2,\ d_{min} \leq z \leq d_{max}\}$. Let $\boldsymbol{R_g} \in \mathbb{R}^{3 \times 3}$ be the rotation matrix derived by the grasp direction of grasp point $\boldsymbol{p_g} \in \mathbb{R}^3$, the total possible grasp region $\tilde{S}$ in the camera frame is formulated as:

$$\tilde{S} = \{\boldsymbol{R_g}\boldsymbol{x} + \boldsymbol{p_g} \mid \boldsymbol{x} \in S,\ \boldsymbol{p_g} \in G\}. \tag{1}$$

We voxelize the possible grasp region $\tilde{S}$ with a fixed voxel size $v$ and get voxel set $\tilde{S}_v$, which is treated as the local occupancy prediction region.

### 3.2 MULTI-GROUP TRI-PLANE

Computation over the entire 3-D scene volume is computationally forbidden for large scenes. To avoid it, we devise a scheme of multi-group tri-plane projection for holistic / local scene context extraction in cluttered scenes. Each group of tri-plane is composed of three feature planes that pool the spatial features projected onto three orthogonal coordinates in some frame. Specifically, we implement the feature on each plane as the aggregation of both point cloud embeddings and the point density along an axis. Importantly, the above process of tri-plane projection is lossy, thus we further propose to use multiple groups of tri-planes that differ in 3-D rotations and share the same origin, thereby more key information can be preserved via diverse aggregations.

To ensure the diversity across different tri-planes, we conduct a spherical linear interpolation of quaternion (Shoemake, 1985) to draw multiple tri-plane coordinate rotations uniformly in the rotation group **SO(3)**. Given the start and the end quaternions $\boldsymbol{q}_1, \boldsymbol{q}_2 \in \mathbb{R}^4$ with $||\boldsymbol{q}_1|| = ||\boldsymbol{q}_2|| = 1$, and the number of tri-plane groups $K$, the interpolated coordinate frame rotations are:

$$\boldsymbol{q}_i = \frac{sin[(1 - \frac{i}{N})\phi]\boldsymbol{q}_1 + sin(\frac{i}{N}\phi)\boldsymbol{q}_2}{sin\phi},\ i = 0, 1, ..., K - 1, \tag{2}$$

where $\phi = arccos(\boldsymbol{q}_1^T \boldsymbol{q}_2)$. Then the quaterion $\boldsymbol{q}_i = (x_i, y_i, z_i, w_i)$ can be transformed to a rotation matrix $\boldsymbol{R}_i \in \mathbb{R}^{3 \times 3}$ by:

$$\boldsymbol{R}_i = \begin{bmatrix} 1 - 2z_i^2 - 2w_i^2 & 2y_i z_i + 2x_i w_i & 2y_i w_i - 2x_i z_i \\ 2y_i z_i - 2x_i w_i & 1 - 2y_i^2 - 2w_i^2 & 2z_i w_i + 2x_i y_i \\ 2y_i w_i + 2x_i z_i & 2z_i w_i - 2x_i y_i & 1 - 2y_i^2 - 2z_i^2 \end{bmatrix}. \tag{3}$$

In practice we set $\boldsymbol{q}_1$ as the identity rotation and $\boldsymbol{q}_2$ to be the rotation that rotates 90° along x-axis first and follows by a second 90° rotation along y-axis to generate diverse tri-plane rotations.

Next, all tri-planes aggregate the point cloud embeddings and the point density along each axis separately. Let $\boldsymbol{F}_{T_{ij}} \in \mathbb{R}^{C_P \times H \times W}$ and $\boldsymbol{D}_{ij} \in \mathbb{N}^{1 \times H \times W}$ be the aggregated point cloud embeddings and the point density on the $i$-th ($i \in \{0, 1, 2\}$) plane of the $j$-th ($j \in \{0, ..., K-1\}$) group respectively, where $H$ and $W$ define the resolution of the plane. The aggregated point cloud embedding and the density located at $(x, y)$ are calculated as:

$$\boldsymbol{F}_{T_{ij}}(x, y) = \mathcal{A}(\{\mathbf{1}_{proj_i}(\boldsymbol{R}_j\boldsymbol{p},\ x,\ y) \cdot \boldsymbol{f_p} \mid \boldsymbol{p} \in \boldsymbol{P}\}),\ \boldsymbol{D}_{ij}(x, y) = \sum_{\boldsymbol{p} \in \boldsymbol{P}} \mathbf{1}_{proj_i}(\boldsymbol{R}_j\boldsymbol{p},\ x,\ y), \tag{4}$$

where $\boldsymbol{f_p} \in \mathbb{R}^{C_P}$ is the $\boldsymbol{p}$'s corresponding embedding in $\boldsymbol{F_P}$, $\mathbf{1}_{proj_i}(\cdot,\ x,\ y)$ is the indicative function showing whether a point's normalized projected point along the $i$-th axis locates at $(x, y)$,

and $\mathcal{A}(\cdot)$ could be any kind of aggregation function (we choose max-pooling). Afterwards, $\boldsymbol{D}_{ij}$ is normalized via soft-max and obtain density $\tilde{\boldsymbol{D}}_{ij} \in [0,1]^{1 \times H \times W}$. Ultimately, three 2-D plane-oriented encoders $\mathcal{E}_i$ ($i \in \{0,1,2\}$) shared by all groups fuse the aggregated embedding and density into multi-group tri-plane context $\tilde{\boldsymbol{F}}_{T_{ij}} \in \mathbb{R}^{C_T \times H \times W}$ by:

$$\tilde{\boldsymbol{F}}_{T_{ij}} = \mathcal{E}_i(\boldsymbol{F}_{T_{ij}} \oplus \tilde{\boldsymbol{D}}_{ij}), \tag{5}$$

where $\oplus$ refers to concatenation and $C_T$ is the number of feature channels of each plane.

The utilization of multi-group tri-plane approximately captures global scene context in a concise way. On the one hand, more aggregation views improve the possibility of restoring features for the occluded parts and enriches the 3-D shape clues. On the other hand, it significantly reduces the data size during calculation and avoids the direct operation on dense 3D volume features. The spatial resolution of tri-planes can thus be set to be larger for better representing delicate shape information.

## 3.3 Local Occupancy Query

We further propose a feature query scheme for efficiently fusing the global and local context of the scene, for the sake of occupancy estimation. The target points to be queried $\boldsymbol{P_q} \in \mathbb{R}^{3 \times M}$ are the centers of the voxels in local occupancy region $\tilde{S}_v$, where $M$ is the number of the voxels in $\tilde{S}_v$. For each queried point $\boldsymbol{p_q} \in \boldsymbol{P_q}$, its global context $\boldsymbol{f}_G$ is the fusion of the bi-linear interpolated features on the projection points of different planes. Specifically, an encoder $\tilde{\mathcal{E}}_1$ shared by all tri-plane groups will first fuse the three interpolated features from $j$-th group into $\boldsymbol{f}_{T_j}$, and an another encoder $\tilde{\mathcal{E}}_2$ will then fuse the features from different groups into $\boldsymbol{f}_G$:

$$\boldsymbol{f}_{ij} = \mathcal{BI}(\tilde{\boldsymbol{F}}_{T_{ij}}, proj_i(\boldsymbol{R}_j \boldsymbol{p_q})), \quad \boldsymbol{f}_{T_j} = \tilde{\mathcal{E}}_1(\bigoplus_i \boldsymbol{f}_{ij}), \quad \boldsymbol{f}_G = \tilde{\mathcal{E}}_2(\bigoplus_j \boldsymbol{f}_{T_j}), \tag{6}$$

where $proj_i(\cdot)$ is the function which calculates the normalized projected point along the $i$-th axis, $\mathcal{BI}(\cdot, \cdot)$ is the bi-linear interpolation function and $\bigoplus$ denotes the concentration. While global context $\boldsymbol{f}_G$ contains the long-distance context related to the quering point, it still needs delicate local shape context to predict occupancy. For this reason, the local context $\boldsymbol{f}_L$ draws the information from observed point clouds and the position embeddings of the relative translation to the nearest grasp point. We first find $\boldsymbol{p_q}$'s nearest neighbour $\boldsymbol{p}'$ in $G$ and the corresponding point cloud embedding $\boldsymbol{f}_{\boldsymbol{p}'}$, then the local context $\boldsymbol{f}_L$ is calculated as:

$$\boldsymbol{f}_L = \boldsymbol{f}_{\boldsymbol{p}'} \oplus \mathcal{E}_{PE}(\boldsymbol{p_q}, \boldsymbol{p}', \boldsymbol{p_q} - \boldsymbol{p}'), \tag{7}$$

where $\mathcal{E}_{PE}$ is an MLP to generate position embedding. At last, the queried feature $\boldsymbol{f}_{\boldsymbol{p_q}} \in \mathbb{R}^{C_Q}$ is obtained by $\boldsymbol{f}_{\boldsymbol{p_q}} = \boldsymbol{f}_G \oplus \boldsymbol{f}_L$ where $C_Q$ is the number of feature channels, and an MLP based decoder predicts the occupancy probability of $\boldsymbol{p_q}$ according to $\boldsymbol{f}_{\boldsymbol{p_q}}$.

## 3.4 Grasp Pose Estimation With Completed Local Shape

Having obtained the completed shape around grasp-related regions, the local occupancy-enhanced grasp pose estimation is done in three sequential steps: extracting occupancy-enhanced local shape feature, refining grasp direction and decoding shape feature into grasp poses.

**Occupancy-enhanced local shape feature extraction**. With the queried features and the occupancy probability of a grasp region, we can extract local occupancy-enhanced feature from completed shape information in local regions. For each grasp point $\boldsymbol{p_g} \in G$, we conduct the cylinder crop as in Fang et al. (2020) and get the predicted occupied voxels in its local grasp region. Assume the center points of occupied voxels in one local grasp region are $\boldsymbol{P}_o \in \mathbb{R}^{3 \times N_o}$ and their corresponding queried features are $\boldsymbol{F}_o \in \mathbb{R}^{C_Q \times N_o}$, where $N_o$ is the number of occupied voxels in local grasp region. Next, as $\boldsymbol{P}_o$ is an explicit form of local shape, a shape encoder composed of 4 point set abstraction layers proposed in Pointnet++ Qi et al. (2017) extracts the delicate shape feature from $\boldsymbol{P}_o$. In addition, some important implicit shape information may have been embedded in $\boldsymbol{F}_o$, therefore we randomly sample a few key points from $\boldsymbol{P}_o$. Their corresponding queried features in $\boldsymbol{F}_o$ are processed with max-pooling as the holistic feature of the local region. Finally, these two kinds of features are concatenated as the local occupancy-enhanced shape feature.

**Grasp direction refinement**. From the experiments, we observe that compared with the other grasp pose parameters, grasp directions have a greater impact on the quality of the grasp poses. However, the grasp directions predicted for local grasp regions previously only consider the information from the incomplete single view point cloud. Therefore, due to the lack of complete scene information, the previously predicted directions may result in bad grasp poses and increases the risk of collision, especially in occluded areas. To address this, we propose to refine the grasp direction based on complete local shape information. To this end, after local occupancy prediction, we extract the local shape feature of the grasp region with the method mentioned above, and then use it to re-estimate the grasp direction. With the complete shape information, a refined grasp direction can be inferred. Then, we extract local shape feature again in the new grasp region for grasp pose estimation. It should be noted that after refining the grasp direction, there is no need of querying occupancy again in the new grasp region. This is because we find the densely sampled grasp points make the grasp region $\tilde{S}_v$ cover almost every occupied voxels around the grasp affordance areas, which means the refined local region rarely contains undiscovered occupied voxels.

**Estimating grasp poses**. Up to now the grasp point and direction have been determined. For the rest parameters of a grasp pose, including in-plane rotation, grasp depth and width, we use a grasp pose decoder with the same structure in Wang et al. (2021) to decode the occupancy-enhanced local shape feature. It regresses the grasp width and the grasp score for several combinations of in-plane rotations and grasp depths. Finally, the parameter combination with the maximum grasp score is regarded as the grasp pose estimation result of each grasp region.

## 3.5 LOSS FUNCTION

Our model is optimized in an end-to-end fashion and supervised by the ground-truth local occupancy labels and grasp pose labels. The loss function is a multi-task loss consisting of local occupancy loss and grasp pose loss. Assume the occupancy prediction in local grasp regions is $\boldsymbol{o}$, the corresponding occupancy ground truth $\boldsymbol{o}_{gt}$ is generated by cropping the total scene occupancy with the predicted local region, then the occupancy loss is a binary cross-entropy loss $L_o(\boldsymbol{o}, \boldsymbol{o}_{gt})$. The grasp pose loss consists of the affordance segmentation loss $L_a$, view-affordance loss $L_v$, grasp width loss $L_w$ and grasp score loss $L_s$. Following common practice, $L_a$ is a binary cross-entropy loss and the rested losses are smooth-L1 losses. Putting all together, assume $gt$ to be the ground-truth, the total loss is written as $L = L_o(\boldsymbol{o}, \boldsymbol{o}_{gt}) + \lambda_1 L_a(\boldsymbol{a}, \boldsymbol{a}_{gt}) + \lambda_2 L_v(\boldsymbol{v}, \boldsymbol{v}_{gt}) + \lambda_3(L_w(\boldsymbol{w}, \boldsymbol{w}_{gt}) + L_s(\boldsymbol{s}, \boldsymbol{s}_{gt}))$, where $\lambda_1, \lambda_2, \lambda_3$ are loss weights and $\boldsymbol{a}, \boldsymbol{v}, \boldsymbol{w}, \boldsymbol{s}$ denote predicted affordance segmentation, view-affordance, grasp width and grasp score respectively.

## 4 EXPERIMENTS

**Dataset**. We evaluate the proposed grasping model on GraspNet-1Billion benchmark (Fang et al., 2020). It is a large-scale real-world grasping dataset containing 190 cluttered grasping scenes and 97,280 RGB-D images captured by 2 kinds of RGB-D cameras from 256 different views. 88 objects with dense grasp pose annotations are provided. The test set is divided into 3 levels (seen / similar / novel) according to the familiarity of objects. For the occupancy label, we utilize the Signed Distance Function (SDF) of each object and the object pose annotations to generate scene level occupancy.

**Baselines**. We run several state-of-the-art competing methods on GraspNet 1Billion (Fang et al., 2020) benchmark, including Morrison et al. (2018); Chu et al. (2018); ten Pas et al. (2017); Liang et al. (2019); Fang et al. (2020); Gou et al. (2021); Qin et al. (2023); Ma & Huang (2022); Wang et al. (2021). For fair comparison, all baseline models only read single-view point clouds as ours.

**Metrics**. For grasp pose estimation, we report the grasp **AP** of the top-50 grasp poses after grasp pose-NMS. The grasp **AP** is the average of $AP_\mu$, where $\mu$ is the friction coefficient ranging from 0.2 to 1.2 and $AP_\mu$ is the average of $Precision@k$ for k ranges from 1 to 50. For the local occupancy prediction, we report the $F_1$-Score and the volumetric IOU in the predicted local occupancy region. $F_1$-Score is the harmonic average of the precision and recall and volumetric IOU is the intersection volume over the union volume for occupancy prediction and ground truth.

**Implementation details**. As the number of voxels varies across different scenes, for the convenience of mini-batch training, we randomly sample 15,000 voxels for occupancy prediction. For the

Table 1: Experimental results on GraspNet-1Billion benchmark for baselines and our proposed model. Since each scenario in GraspNet-1Billion was captured by two kinds of devices (RealSense or Kinect), for each experiment the two scores correspond the results for RealSense / Kinect respectively. **Avg.** means average, **CD** is collision detection and - means not being reported. Best scores are displayed in red.

| Method | Seen | | | Similar | | | Novel | | | Avg. |
|---|---|---|---|---|---|---|---|---|---|---|
| | **AP** | $AP_{0.8}$ | $AP_{0.4}$ | **AP** | $AP_{0.8}$ | $AP_{0.4}$ | **AP** | $AP_{0.8}$ | $AP_{0.4}$ | **AP** |
| **w/o CD** | | | | | | | | | | |
| Morrison et al. (2018) | 15.48/16.89 | 21.84/22.47 | 10.25/11.23 | 13.26/15.05 | 18.37/19.76 | 4.62/6.19 | 5.52/7.38 | 5.93/8.78 | 1.86/1.32 | 11.42/13.11 |
| Chu et al. (2018) | 15.97/17.59 | 23.66/24.67 | 10.80/12.47 | 15.41/17.36 | 20.21/21.64 | 7.06/8.86 | 7.64/8.04 | 8.69/9.34 | 2.52/1.76 | 13.01/14.33 |
| ten Pas et al. (2017) | 22.87/24.38 | 28.53/30.06 | 12.84/13.46 | 21.33/23.18 | 27.83/28.64 | 9.64/11.32 | 8.24/9.58 | 8.89/10.14 | 2.67/3.16 | 17.48/19.05 |
| Liang et al. (2019) | 25.96/27.59 | 33.01/34.21 | 15.37/17.83 | 22.68/24.38 | 29.15/30.84 | 10.76/12.83 | 9.23/10.66 | 9.89/11.24 | 2.74/3.21 | 19.29/20.88 |
| Fang et al. (2020) | 27.56/29.88 | 33.43/36.19 | 16.95/19.31 | 26.11/27.84 | 34.18/33.19 | 14.23/16.62 | 10.55/11.51 | 11.25/12.92 | 3.98/3.56 | 21.41/23.08 |
| Gou et al. (2021) | 27.98/32.08 | 33.47/39.46 | 17.75/20.85 | 27.23/30.40 | 36.34/37.87 | 15.60/18.72 | 12.25/13.08 | 12.45/13.79 | 5.62/6.01 | 22.49/25.19 |
| Qin et al. (2023) | 49.85/47.32 | 59.67/57.27 | 42.24/38.55 | 41.46/35.73 | 50.31/44.22 | 33.69/26.99 | 17.48/16.10 | 21.83/20.01 | 7.90/7.81 | 36.26/33.05 |
| Ma & Huang (2022) | 58.95/ - | 68.18/ - | 54.88/ - | 52.97/ - | 63.24/ - | 46.99/ - | 22.63/ - | 28.53/ - | 12.00/ - | 44.85/ - |
| Wang et al. (2021) | 65.70/61.19 | 76.25/71.46 | 61.08/56.04 | 53.75/47.39 | 65.04/56.78 | 45.97/40.43 | 23.98/19.01 | 29.93/23.73 | 14.05/10.60 | 47.81/42.53 |
| Ours | 70.85/62.99 | 82.21/73.47 | 65.69/56.39 | 63.37/52.92 | 76.33/63.52 | 55.21/45.47 | 27.29/20.79 | 36.25/25.74 | 14.34/12.13 | 53.84/45.60 |
| **w/ CD** | | | | | | | | | | |
| Qin et al. (2023) + CD | 52.16 / 50.45 | 62.71/61.22 | 43.14/40.64 | 44.69/38.62 | 54.52/48.75 | 35.37/28.81 | 19.26/17.66 | 23.93/21.94 | 8.89/8.29 | 38.70/35.58 |
| Ma & Huang (2022) + CD | 63.83/ - | 74.25/ - | 58.66/ - | 58.46/ - | 70.05/ - | 51.32/ - | 24.63/ - | 31.05/ - | 12.85/ - | 48.97/ - |
| Wang et al. (2021) + CD | 67.12/63.50 | 78.46/74.54 | 60.90/58.11 | 54.81/49.18 | 66.72/59.27 | 46.17/41.98 | 24.31/19.78 | 30.52/24.60 | 14.23/11.17 | 48.75/44.15 |
| Ours + CD | 72.89 /64.82 | 85.02/75.61 | 66.64/57.70 | 65.29/55.22 | 79.02/66.47 | 56.06/46.62 | 28.37/20.90 | 35.62/25.89 | 14.61/11.58 | 55.52/46.89 |

inference period, there is no such restriction about the number of voxels. In addition, the view-wise affordance loss is the averaged of the view-wise affordance before and after grasp direction refinement. The width loss is only calculated when the corresponding grasp score is positive. As for the occupancy ground truth label, it is generated from the object models in Fang et al. (2020). We first generate object-level occupancy label by the signed distance function (SDF) of each object. We sample points with a fixed voxel size in the self coordinate frame of the object, and for each point $x \in \mathbb{R}^3$, $SDF(x) \leq 0$ means $x$ is an inner point. These inner points are regarded as the centers of occupied voxels. Then both of the object and table plane voxels are transformed to the camera frame to generate the scene-level occupancy labels. The object poses in camera frame is annotated by the dataset providers.

The default number of tri-plane groups is set to be 3 and the default tri-plane size $H \times W$ is set as $64 \times 64$. The 3D UNet backbone has 14 layers with the output dimension $C_P = 256$. The tri-plane encoders $\{\mathcal{E}_i\}$ are 6-layer ResNets with the output dimension $C_T = 128$. For the local grasp regions, the gripper radius $r$ is 0.05m, $d_{min}$ and $d_{max}$ are -0.01m and 0.04m, and the voxel size $v$ is 0.01m. The number of the queried features' channels is $C_Q = 512$. For each grasp point, we predict grasp scores and widths for 12 possible in-plane rotations combined with 4 possible depths. We train our model on 2 Nvidia TITAN X GPUs for 12 epochs with the Adam optimizer (Kingma & Ba, 2015). The learning rate is 0.001 and the batch size is 4. The loss weights are $\lambda_1 = 10$, $\lambda_2 = 100$, $\lambda_3 = 10$ without further empirical finetuning.

**Experimental results**. The grasping performance on RealSense and Kinect RGB-D cameras are reported in Table 1. Note that Ma & Huang (2022) uses different grasp points sampling strategies so that it performs better in some cases (*e.g.*, $AP_{0.4}$ of similar objects). Under the metric of **AP**, our method achieves scores of 70.85 / 63.37 / 27.29 on seen / similar / novel scenes using the sensor of RealSense respectively. This outperforms previous state-of-the-art method by large margins of 5.15 / 9.62 / 3.31. Similar observation holds for the Kinect-captured data. Notice that the promotion of the similar scenes is the most distinct among three test levels. We give this credit to that the grasp pose estimation with only incompletely observed shape has a weakness of generalizing to other objects due to the lack of shape information. With the local occupancy enhancement, grasping module can associate grasp poses with the completed shape and thus improves the ability to be generalized to the objects with similar shapes. Moreover, as local occupancy prediction can reconstruct the scene with an explicit voxel representation, utilizing the predicted occupancy also improves the effectiveness of collision detection. With an additional post processing to filter out the grasp poses collided with predicted occupancy, the results are improved by 1.68 **AP** on average.

To prove the effectiveness of local occupancy prediction, we report the performance of occupancy prediction in the grasp regions in Table 2. It turns out that the local occupancy prediction is capable to complete the shape of objects in grasp regions and can be generalized to different scenes.

Next, a comparison of the performance and efficiency between different occupancy prediction strategies (on RealSense set) is shown in Table 3. The grasp **AP**, volumetric IOU, inference time and number of calculated voxels are reported. We compare four kinds of strategies with our method 'Tri-

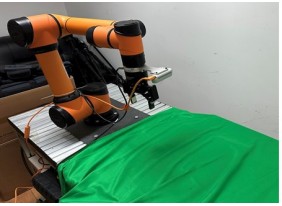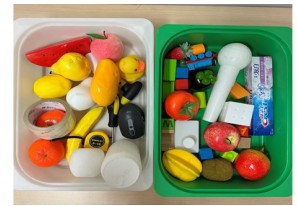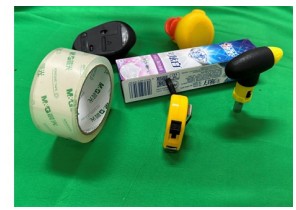

Figure 3: Settings of real-world experiments. *Left*: the configuration of the grasping system. *Middle*: objects used for grasping. *Right*: an example of the grasping scene.

Table 2: Local occupancy estimation performances, where RS, KN stand for RealSense, Kinect respectively.

|  |  | Seen | Smilar | Novel | Avg. |
|---|---|---|---|---|---|
| F$_1$-Score | RS | 0.787 | 0.764 | 0.708 | 0.754 |
|  | KN | 0.779 | 0.763 | 0.709 | 0.750 |
| Volumetric IOU | RS | 0.681 | 0.645 | 0.589 | 0.638 |
|  | KN | 0.683 | 0.636 | 0.586 | 0.635 |

Table 3: Comparison of different strategiesin occupancy estimation.

| Setting | AP | IOU | Time | Voxels |
|---|---|---|---|---|
| w/o Occupancy | 49.72 | - | **0.13s** | - |
| 3D Conv | 51.02 | 0.567 | 0.39s | 216000 |
| Tri-plane Global | 51.75 | 0.596 | 0.35s | 216000 |
| Ball Query | 50.60 | 0.449 | 0.16s | ∼ 14000 |
| Tri-plane Local | **53.84** | **0.638** | 0.18s | ∼ 14000 |

Table 4: Ablation study of different modules. **VR** is short for 'view refinement'.

| $F_{T_{ij}}$ | $D_{ij}$ | $f_L$ | VR | AP | IOU | Time |
|---|---|---|---|---|---|---|
| ✗ | ✓ | ✓ | ✓ | 53.03 | 0.612 | 0.16s |
| ✓ | ✗ | ✓ | ✓ | 53.33 | 0.626 | 0.18s |
| ✓ | ✓ | ✗ | ✓ | 52.40 | 0.569 | 0.16s |
| ✓ | ✓ | ✓ | ✗ | 53.61 | 0.638 | 0.18s |
| ✓ | ✓ | ✓ | ✓ | 53.84 | 0.638 | 0.18s |

Table 5: Performance of real-world grasping. Please refer to the main text for more explanations.

| Method | Objects | Attempts | Success Rate |
|---|---|---|---|
| Baseline | 50 | 57 | 87.72% |
| Ours | 50 | 53 | **94.34%** |

plane Local'. The first one is the baseline without occupancy enhancement. The second type '3D Conv' stands for calculating dense 3D volume features with 3D convolutions proposed in Li et al. (2020) and predicting occupancy in a predefined region with a resolution of $60^3$. The third kind 'Tri-plane Global' denotes to use multi-group tri-plane to capture scene context but to predict occupancy in the predefined region. The forth kind 'Ball Query' predicts occupancy in local occupancy region, but only uses the features from nearby point cloud embedding by conducting ball query proposed in Qi et al. (2017) to get local context. Note that the '3D Conv' and 'Tri-plane Global' models cost too large GPU memory for end-to-end training, thus we first train occupancy prediction modules and then freeze them during training grasp pose estimator. The result shows that predicting occupancy within the local grasp regions and aggregating scene context with multi-group tri-plane are important for reducing the computational cost without losing the performance. Theoretically, the complexity of multi-group tri-plane aggregation is $O(KHW)$ and the complexity of local occupancy query is $O(KM)$, while the complexity of learning 3D feature volume is $O(HWD)$, where $D$ is the depth length. Since the area of grasp regions is far less than the whole scene (statistically 15 times in Table 3), $K(HW + M)$ is smaller than $HWD$ by a non-trivial factor. Therefore the overhead of occupancy prediction is greatly lightened and the additional inference time is acceptable for real-time practical application. Besides, demonstrated by the poor performance of 'Ball Query', capturing long-distance scene context is crucial for occupancy prediction in cluttered scenes. These results prove the effectiveness as well as the efficiency of our method.

**Ablation Study**. We explore the effect of each design through ablation studies. The default set up is $K = 3, H = W = 64$. First we compare the default setting with different modules combinations in Table 4. From the first two rows, aggregating point cloud embedding and point density are both helpful for occupancy prediction and grasp pose estimation. The third row shows that local context in occupancy query is necessary for delicate shape information. The forth row shows the effectiveness of refining view with completed shape feature. All of these are combined to achieve the highest performance. Moreover, we compare the performance of different multi-group tri-plane settings. In specific, $K = 1, 3, 5$ and $H = W = 64, 128, 256$ are evaluated respectively. The results show that

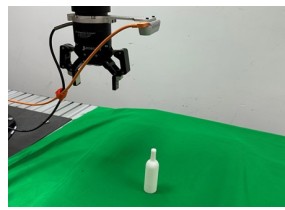 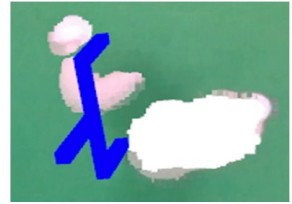 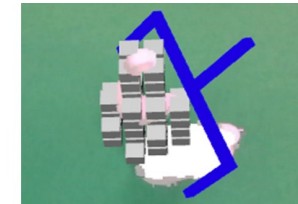

Figure 4: Comparison in real-world test. *Left*: a tiny bottle is right below the camera so the observation is severely self-occluded. *Middle*: due to the lack of complete shape, the grasp pose estimated by the baseline collides with the bottle. *Right*: Our method reconstructs the complete shape of the grasp region and succeeds.

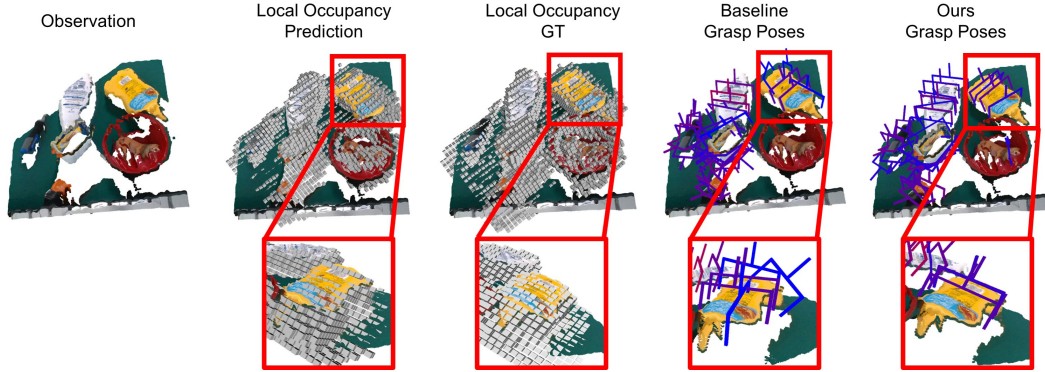

Figure 5: Visualization of the predicted local occupancy and grasp poses of our method. Some grasp poses predicted by the baseline without occupancy enhancement fails in grasping the yellow bottle. More visualization examples can be found in the Appendix.

with the number of tri-plane groups growing, the volumetric IOU and the grasp **AP** both increase (*e.g.*, AP is elevated from 53.14 to 53.89 when K is increased from 1 to 5 with $H = W = 64$). Our model chooses $K = 3$ groups for the balance of computational cost and the performance. However, we find that higher resolutions of plane is not obviously beneficial to the performance, only noting trivial changes by varying $H, W$. We regard this is because the scale of grasping scene in the dataset is not very large, therefore a lower resolution is sufficient to capture enough scene context.

**Real-world grasping**. To examine our model's ability in practical application, we conduct a real-world visual grasping experiment. As shown in Figure 3, we use an Aubo-i5 robotic arm with an Intel RealSense D435i RGB-D camera mounted on it to capture point cloud observations. The gripper is a Robotiq 2F-85 two-finger gripper. The experiment requires the visual grasping system to estimate grasp poses in cluttered scenes with 5-8 objects on the table and pick them up. We compare the proposed model with the baseline without occupancy enhancement. This experiment shows that our method surpass the baseline by 6.62% (see Table 5). Visualization comparisons between the baseline and our method are shown in Figures 4 and 5. They provide clear evidence for the importance of reconstructing complete shape in object grasping.

## 5 CONCLUSION

In this paper, we propose a local occupancy-enhanced grasp pose estimation method. By completing the missing shape information in the candidate grasp regions from a single view observation, our method boosts the performance of object grasping with the enhanced local shape feature. Besides, to infer local occupancy efficiently and effectively, the multi-group tri-plane is presented to capture long-distance scene context as well as preserving 3D information from diverse aggregation views. Comprehensive experiments on benchmarks and real robotic arm demonstrate that completed shape context is essential to grasp pose estimation in cluttered scenes, and our local occupancy prediction is of significance for promoting the performance of object grasping.

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

APPENDIX: VISUALIZATION

Visualization examples of predicted local occupancy and grasp poses are shown in Figures 6 and 7.

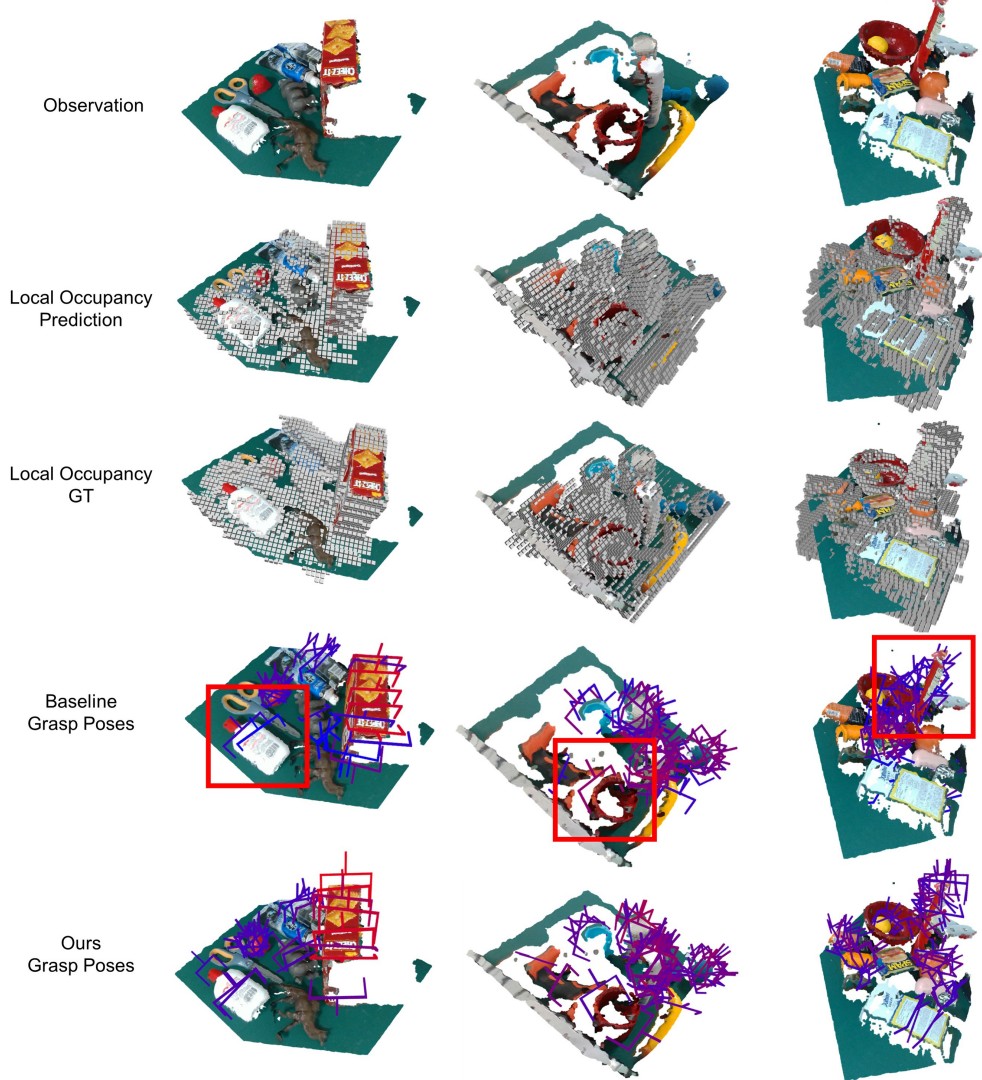

Figure 6: Visualizations of predicted local occupancy and grasp poses. Bad grasp poses proposed by the baseline are marked with red boxes.

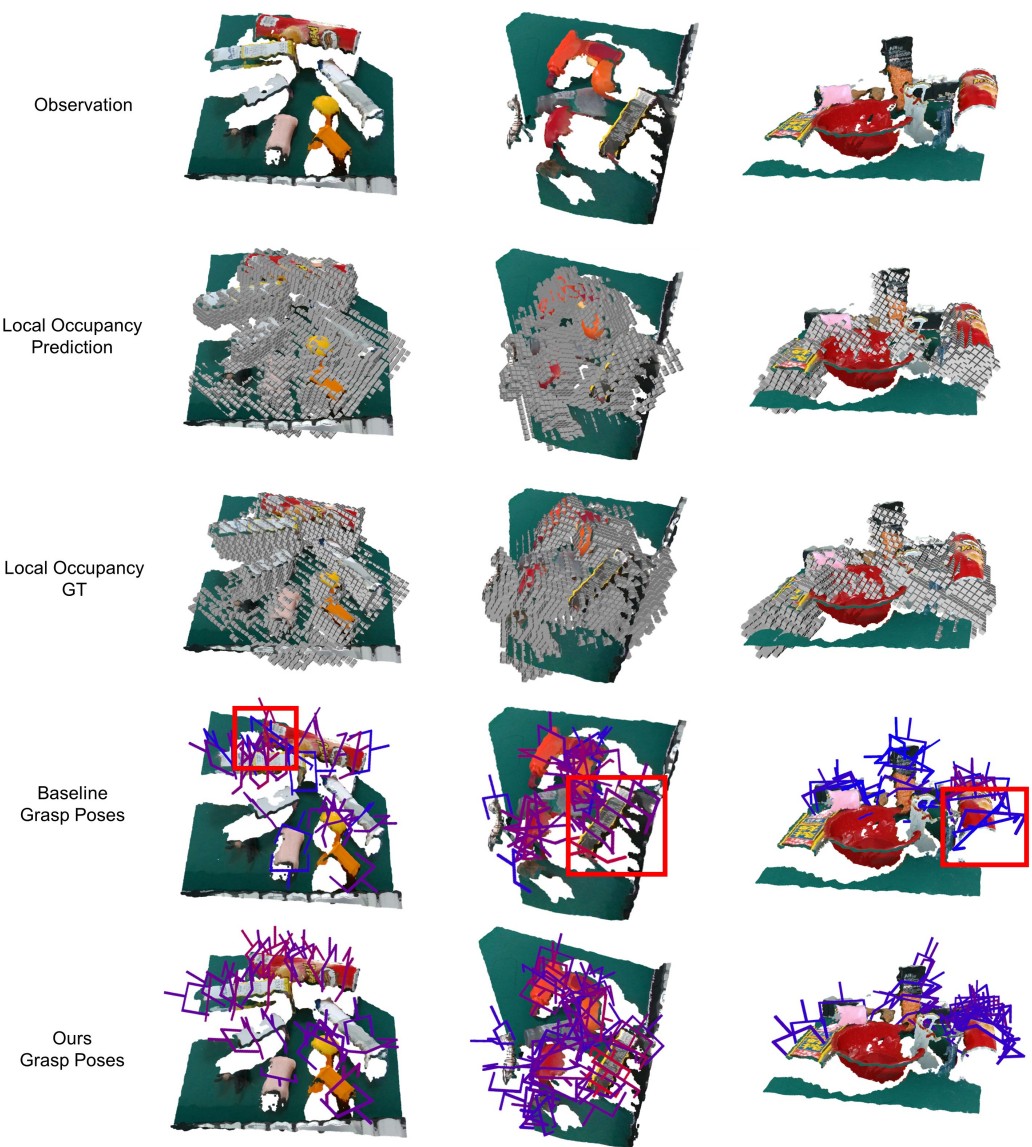

Figure 7: Visualizations of predicted local occupancy and grasp poses. Bad grasp poses proposed by the baseline are marked with red boxes.

