# OpenReview forum: "Multi-Group Tri-plane Based Local Occupancy Estimation for Object Grasping"
_ICLR.cc/2024/Conference — ICLR 2024 Conference Withdrawn Submission_

### Official Review · Reviewer_zrC8 · 2023-10-23

**Soundness:** 3 good
**Presentation:** 2 fair
**Contribution:** 2 fair
**Rating:** 5
**Confidence:** 4

**Summary:**

This paper proposes a grasp detection method, where a multi-group tri-plane scheme is used to predict the occupancy and then 6D grasp poses are estimated using the occupancy prediction. Experimental results on large-scale datasets and real robot grasping show the effectiveness of the proposed method.

**Strengths:**

The strength of this paper is that it outperforms existing methods on the GraspNet-1B benchmark and shows its effectiveness for real robot grasping.

**Weaknesses:**

The main weaknesses of the paper is that the novelty is limited. As described in the related work, the tri-plane representations have been adopted in many existing works. The improvement by using multi-group tri-plane representations is incremental in my opinion.In the experiment section, the authors compare the effect of different numbers of tri-plane, and the improvement from 53.14 to 53.89 is not significant enough.

An experiment on how to choose the tri-plane frame and how it affects the results would be interesting.

**Questions:**

1. The principle of the grasp affordance segmentation procedure used in the paper should be introduced. What's ``the view-wise affordance''?
2. The table of comparison results of different number of tri-plane group and resolution is not included in the paper.
3. Failure cases and analysis can be added.

---

### Official Review · Reviewer_DkPo · 2023-10-31

**Soundness:** 3 good
**Presentation:** 3 good
**Contribution:** 3 good
**Rating:** 6
**Confidence:** 3

**Summary:**

This paper proposes a method to estimate grasp points from point cloud data. An input point cloud is queried for candidate grasps, aided a tri-plane based local occupancy estimation that essentially locally completes the unobserved shape of the objects in the scene. Experimental results demonstrate the validity of the proposed approach, and an ablation study justifies and discusses the design choices.

**Strengths:**

The experimental results appear convincing, achieving SOtA results. Additionally, this work effectively bypasses the object segmentation problem, and solves the 3D object segmentation problem in the same pipeline as grasp proposal. More strengths include a rather clear presentation (although please fix numerous small english errors throughout the paper) and the immense applicability of the tackled problem.

**Weaknesses:**

In my opinion, slerp is fairly standard, as is the conversion between quaternions and rotation matrices. consider replacing equations (2) and (3) with citations, perhaps to free up space for more technical description of your work if/where needed.

After describing slerp and quaternion transformations, you proceed to use specific rotations to build the proposed descriptor. why specify the rotations? their quaternion parameters could be just randomized, and/or learned. is this specific choice defined/constrained by the nature of the point cloud?

Regarding the choice of max-pooling, how about other choices? Especially given the nature of the proposed descriptor, it might be worth examining the implications of this choice on the final feature representations and how sensitive the results are to this specific aggregation function.


Finally, numerous minor english errors hinder readability (eg facility instead of facilitate, interested local occupancy regions instead of local occupancy regions of interest etc), please have a careful look and fix those.

**Questions:**

Feel free to comment on/refute the weaknesses of the work as I describe them above.

Some more minor issues/comments:

In the literature overview of the "occupancy network" approaches, this category is dismissed as "these methods need to predict occupancy densely for the whole scene, which is unnecessary and overburdened for grasping tasks." This is a strong claim, and although I appreciate cutting down the computational cost of to the proposed approach, one might argue that these might work just as well -- or even better if accuracy is all that is needed. Please comment on/address this issue.

Connected to the issue above, in Section 3.2, "Computation over the entire 3-D scene volume is computationally forbidden for large scenes" (should read 'prohibitive' btw), can you provide some figures to back this claim? I fully believe it will be much more computationally costly, but it would be useful to roughly estimate the difference.

How about the force of the gripper? Is it assumed that the gripper handles hard, rigid objects?

Overall I feel this is a strong paper that is hindered by specific design choices. It is quite possible that I didn't understand all design choices, I'm happy to revise my score if the authors convincingly reply my comments.

---

### Official Review · Reviewer_PGYL · 2023-10-31

**Soundness:** 2 fair
**Presentation:** 1 poor
**Contribution:** 2 fair
**Rating:** 5
**Confidence:** 3

**Summary:**

The paper introduces a new approach for grasping unknown objects from single view by using local occupancy estimation. Grasping of unknown objects from a single view is a challenging problem, as self-occlusion can lead to a poor and fragmentary estimate of the object's shape which can in turn elicit suboptimal grasping strategies. This article proposes to address this limitation by inferring the object's full 3D shape (or occupancy), leading to more robust grasping strategies. The occupancy is learned using multiple tri-plane features to infer local occupied voxels. This improved occupancy lead to better grasp pose estimations, and more successful grasps.


Limitations:
- The paper generally lacks in clarity. It is a good idea to provide a system diagram in Figure 1, but it is very unclear how the different steps relate to the different parts in the method description. It is unclear how the voxel query operation described in the text links the "multi-group tri-plane" to the Global Context" in the diagram. Similarly the box for implicit features and local shape feature is not very informative. Several functions are defined as MLPs, but the paper does not provide more specific information about the architecture.
- Similarly, very little information is provided about how the model's components are trained, or validated (on what data? end to end? for how long?).
- The experimental results seem to compare well with the chosen baselines, but all results lack any form of variance or confidence intervals. It is therefore unclear from those results what conclusion to draw from the ablation study in Table 4: disabling one component seems to only cause a minor decrease in performance.
- Many typos and instances of loose or unclear language remain in the paper, making the argument hard to follow in parts.

Motivation: The argument for the approach with respect to the literature, and novelty is not made convincingly.  The introduction cites recent approaches for general object grasping (eg, Xu et al 2023), but do not discuss them in the remainder of the paper. The novelty of the approach with respect to previous published works and contributions are not clearly stated. Part of the argument is made with respect to the computational efficiency of the coarse/voxel-based occupancy approach, but the evaluation is only done with respect to accuracy of the grasping with respects to traditional methods.

Fundamentally, one would assume that local occupancy prediction of unknown objects from single view implies learning some generic priors about shapes. How well those priors generalise from objects in the training set to other objects is an important question, but it is only provided as aggregated over all objects in the dataset in Table 2, again without variance information (what are the objects where the approach generalises best and worse?). Similarly, the comparison of completion methods in Table 3 only records IOUs in average over the three sets, a breakdown between the three categories would be important here.

In summary, the approach described in the paper appears to work well, but is poorly motivated and analysed. As it stands the paper reads as more of an engineering achievement, which would be more suited to a robotics conference rather than a theoretical contribution.

**Strengths:**

- The problem addressed by the paper, grasping unknown objects from a single view is a relevant and interesting problem.
- To my knowledge, the proposed approach is novel: although it draws from pre-existing components, the combination is distinct enough from existing approaches.
- The paper's experimental results seem good and the model performs favourably with respect to the considered baselines.
- Good to provide an ablation study.
- Good to provide results on real-world experiments

**Weaknesses:**

- The paper generally lacks in clarity. It is a good idea to provide a system diagram in Figure 1, but it is very unclear how the different steps relate to the different parts in the method description. It is unclear how the voxel query operation described in the text links the "multi-group tri-plane" to the Global Context" in the diagram. Similarly the box for implicit features and local shape feature is not very informative. Several functions are defined as MLPs, but the paper does not provide more specific information about the architecture.
- Similarly, very little information is provided about how the model's components are trained, or validated (on what data? end to end? for how long?).
- The experimental results seem to compare well with the chosen baselines, but all results lack any form of variance or confidence intervals. It is therefore unclear from those results what conclusion to draw from the ablation study in Table 4: disabling one component seems to only cause a minor decrease in performance.
- Many typos and instances of loose or unclear language remain in the paper, making the argument hard to follow in parts.

The motivation for the approach with respect to the literature and its novelty lacks clarity.  The introduction cites recent approaches for general object grasping (eg, Xu et al 2023), but do not discuss them in the remainder of the paper. Similarly, part of the argument for the method is made with respect to the computational efficiency of the coarse/voxel-based occupancy approach versus other occupancy methods, but the evaluation is only done with respect to the accuracy of the grasping. More broadly, the novelty of the approach with respect to previous published works and contributions is not clearly stated.

**Questions:**

One would assume that local occupancy prediction of unknown objects from single view implies learning some generic priors about shapes. How well do those priors generalise from objects in the training set to other objects? Table 2 provides only averages, but what are the objects/shapes where the approach generalised best and worse? Similarly, the comparison of completion methods in Table 3 only records IOUs in average over the three sets, what is the breakdown between seen and unseen objects?

Also, what is the stability of the results over different seeds or training sets?

---

### Official Review · Reviewer_pApf · 2023-11-03

**Soundness:** 3 good
**Presentation:** 2 fair
**Contribution:** 3 good
**Rating:** 6
**Confidence:** 4

**Summary:**

This paper proposes an algorithm for 2-finger grasp detection from an input single-view pointcloud. First, regions in the pointcloud with high grasp affordance are identified using an off-the-shelf method. Next, features describing the local context are derived by projecting the point cloud on multiple groups of 3 orthogonal planes, and aggregating point embedding and density. This supports querying features at grid locations around the grasp point. Thus, the single-view shape (which suffers from self occlusion and incompleteness) is "completed" in the feature space. This completed representation is used to refine the grasp direction and to predict other grasp parameters like grasp width.

Experiments on the GraspNet 1Billlion dataset and on the real robot show significant improvements over baselines.

**Strengths:**

- Strong experimental evaluation and improvement over baselines.
- Various design decisions like aggregating point density, including local context, etc. are verified through an ablation study (Table 4).
- Grasping incompletely perceived shapes is an important practical problem e.g. in mobile manipulation in environments not fitted with third-person cameras. Thus, the paper makes meaningful progress in an impactful area.

**Weaknesses:**

- The paper can benefit from more direct comparisons to scene/shape completion methods for grasp detection, especially Lundell et al 2020. If they are object-centric rather than using context from the entire scene, some heuristic methods can be used to convert object completion to scene completion e.g. apply object completion to each segmented object.
- The impact of the initial local occupancy / grasp affordance regression module (Section 3.1) is not investigated in the ablation studies.

**Questions:**

Please perform direct comparisons to scene/shape completion methods for grasp detection.